# *Escherichia marmotae*—a Human Pathogen Easily Misidentified as *Escherichia coli*

Audun Sivertsen,[a] Ruben Dyrhovden,[a] Marit Gjerde Tellevik,[a] Torbjørn Sæle Bruvold,[a] Eirik Nybakken,[a] Dag Harald Skutlaberg,[a] Ingerid Skarstein,[a] Øyvind Kommedal[a,b]

[a]Department of Microbiology, Haukeland University Hospital, Bergen, Norway
[b]Department of Clinical Science, University of Bergen, Bergen, Norway

**ABSTRACT** We hereby present the first descriptions of human-invasive infections caused by *Escherichia marmotae*, a recently described species that encompasses the former *"Escherichia* cryptic clade V." We describe four cases, one acute sepsis of unknown origin, one postoperative sepsis after cholecystectomy, one spondylodiscitis, and one upper urinary tract infection. Cases were identified through unsystematic queries in a single clinical lab over 6 months. Through genome sequencing of the causative strains combined with available genomes from elsewhere, we demonstrate *Es. marmotae* to be a likely ubiquitous species containing genotypic virulence traits associated with *Escherichia* pathogenicity. The invasive isolates were scattered among isolates from a range of nonhuman sources in the phylogenetic analyses, thus indicating inherent virulence in multiple lineages. Pan genome analyses indicate that *Es. marmotae* has a large accessory genome and is likely to obtain ecologically advantageous traits, such as genes encoding antimicrobial resistance. Reliable identification might be possible by matrix-assisted laser desorption ionization–time of flight mass spectrometry (MALDI-TOF MS), but relevant spectra are missing in commercial databases. It can be identified through 16S rRNA gene sequencing. *Escherichia marmotae* could represent a relatively common human pathogen, and improved diagnostics will provide a better understanding of its clinical importance.

**IMPORTANCE** *Escherichia coli* is the most common pathogen found in blood cultures and urine and among the most important pathogenic species in the realm of human health. The notion that some of these isolates are not *Es. coli* but rather another species within the same genus may have implications for what *Es. coli* constitutes. We only recently have obtained methods to separate the two species, which means that possible differences in important clinical aspects, such as antimicrobial resistance rates, virulence, and phylogenetic structure, may exist. We believe that *Es. marmotae* as a common pathogen is new merely because we have not looked or bothered to distinguish between the thousands of invasive *Escherichia* passing through microbiological laboratories each day.

**KEYWORDS** 16S RNA, *Escherichia*, genome analysis, genotypic identification, pathogens, phenotypic identification, phylogenetic analysis, virulence determinants

The type strain of *Escherichia marmotae* was isolated from the gut of the Himalayan marmot (*Marmota himalayana*) and was described as a novel species in 2015 (1). *Es. marmotae* encompasses the environmental *Escherichia* species previously referred to as *"Escherichia* cryptic clade V." Recently, *Es. marmotae* was predicted to represent a potential human pathogen by *in vitro* infection assays and virulence gene characterization (2). The presence of virulence factors may be linked to virulent *Es. coli* occupying the same environmental niche in the vertebrate gut, and it was also suggested that the *astA* gene encoding the heat-stable enterotoxin 1 in enteroaggregative *Es. coli*

Address correspondence to Audun Sivertsen, audun.sivertsen@helse-bergen.no.

The authors declare no conflict of interest.

originates from *Es. marmotae* (3). However, until now, *Es. marmotae* has not been reported from human nonenteric samples. Historically the "*Escherichia* cryptic clade V" has been considered an environmental species (4, 5), a rare finding in human enteric samples and of low pathogenic potential (6, 7).

Here, we present four human clinical cases where *Es. marmotae* was isolated as the likely pathogen of invasive human infections. The cases represent epidemiologically unrelated community-acquired infections and were all discovered in 2021 over a period of 6 months. We also characterize the isolates in terms of antimicrobial resistance and virulence using whole-genome sequencing and investigate the phylogenetic relationships between our human-invasive strains and the 36 *Es. marmotae* strains with complete or draft whole genomes available in NCBI GenBank as well as other strains representing the diversity within the *Escherichia* genus.

## RESULTS AND DISCUSSION

In our lab, species identification of cultured isolates is routinely done by matrix-assisted laser desorption ionization–time of flight mass spectrometry (MALDI-TOF MS). All four cases were discovered by partial 16S rRNA gene sequencing of isolates that obtained an unusually poor (although still valid) score for *Es. coli* by MALDI-TOF MS (range of 1.9 to 2.1 versus normally around 2.3). The sequences shared only 98.8% homology with *Es. coli* but were identified as *Es. marmotae* with 100% identity. The observation period was from January to July 2021.

***Escherichia marmotae* as cause of septicemia, cholangitis, pyelonephritis, and spondylodiscitis. (i) Case 1.** An 80-year-old male with acute myelogenous leukemia and pancytopenia was diagnosed with thoracic spondylodiscitis. A computed tomography (CT)-guided aspiration from this focus provided several milliliters of purulent material from which *Es. marmotae* was recovered in pure culture (isolate HUSEmarmC1). During the stay, the patient also had an episode with *Staphylococcus aureus* bacteremia, but *Sa. aureus* was not detected in the sample from the spondylodiscitis neither by culture nor by an *Sa. aureus*-specific PCR. The spondylodiscitis was successfully treated with an expanded-spectrum cephalosporin followed by a course of per oral ciprofloxacin.

**(ii) Case 2.** An individual presented with a case of pyelonephritis. *Escherichia marmotae* was found in pure culture at $>10^5$ CFU/mL in urine (isolate HUSEmarmC2). No further clinical information was permitted.

**(iii) Case 3.** An 85-year-old previously healthy male was admitted with acute sepsis of unknown origin. In the days prior to this, he had been fertilizing fruit trees with sheep manure. *Escherichia marmotae* was recovered in two pairs of aerobe and anaerobe blood culture bottles (isolate HUSEmarmC3). The patient developed septic shock with multiple organ failure and was transferred to the intensive care unit while treated with an expanded-spectrum cephalosporin. He recovered well and was dismissed from the hospital to a short-term care facility.

**(iv) Case 4.** A 66-year-old male with a history of gallstone disease and well-regulated diabetes type II developed acute postoperative sepsis with multiple organ failure after cholecystectomy a few days earlier. *Es. marmotae* was recovered as the sole agent from two pairs of aerobe and anaerobe blood culture bottles collected upon admission (isolate HUSEmarmC4a). In addition, *Es. marmotae* (isolate HUSEmarmC4b) together with *Streptococcus parasanguinis* and *Enterococcus faecium* were cultured from pus collected from the bile ducts through endoscopic retrograde cholangiopancreatography. The patient responded well to intravenous piperacillin-tazobactam followed by a course of oral trimethoprim-sulfamethoxazole after discharge.

**Whole-genome comparisons show that *Es. marmotae* is monophyletic.** Whole-genome sequencing of the five clinical isolates was performed as described in Materials and Methods. In order to obtain a representative phylogenetic overview of the *Escherichia* genus, we downloaded all 36 additional assemblies identified as *Es. marmotae* available in GenBank together with available assemblies representing the main clades, as shown in Denamur et al. (6). Genome accession numbers as well as relevant metadata are provided in Table S1 in the supplemental material. All genomes

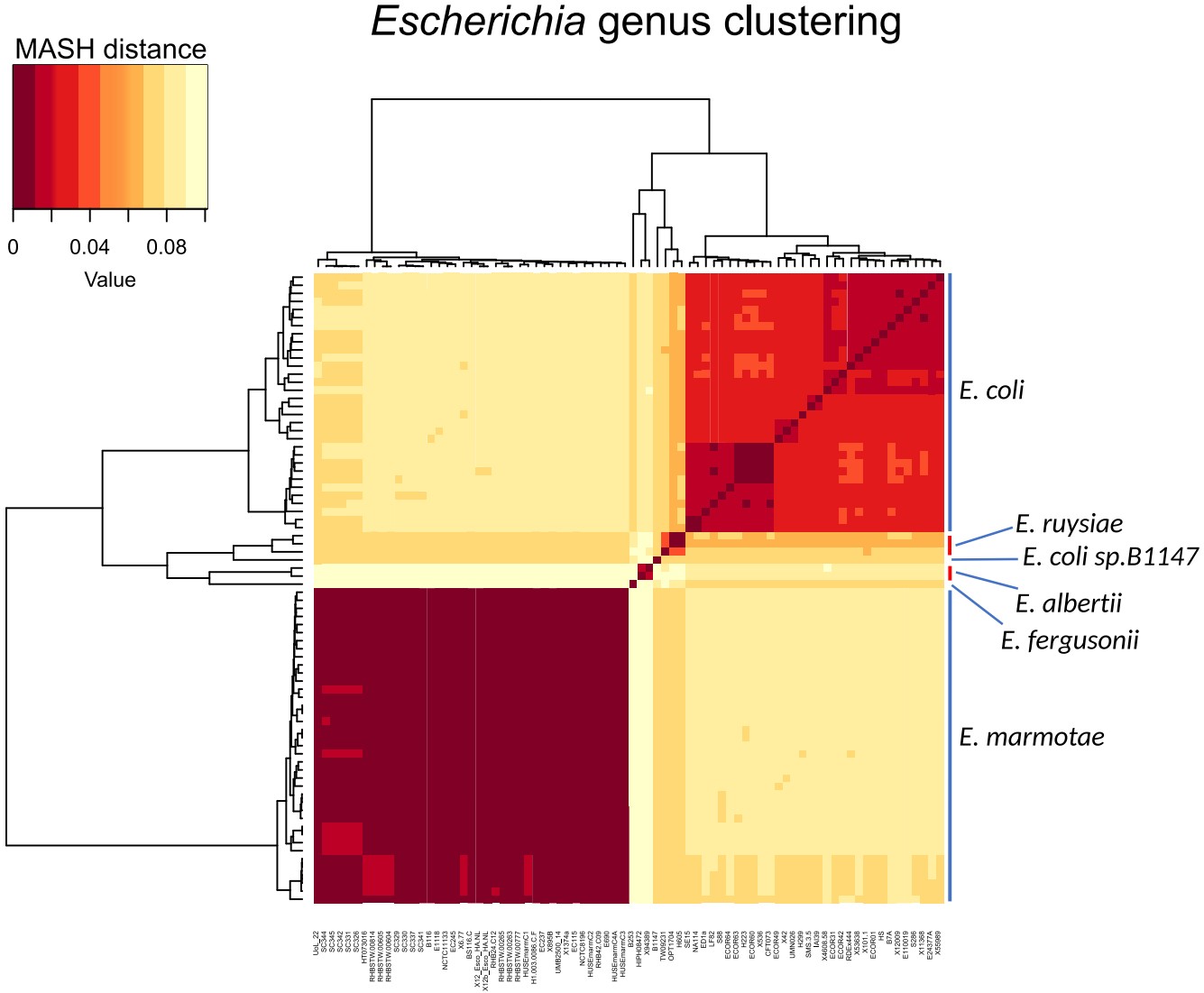

**FIG 1** Clustering of *Escherichia* strains based on small MASH distance (high genetic similarity). Two identical clusterograms surround a heat map showing pair-to-pair comparisons of all *Escherichia* strains, with dark red denoting close pairwise similarity. Apparent blocks represent clusters that correspond to species, as described to the right. Strain names are given at the bottom.

were clustered based on pairwise distance using MASH (8). The resulting clusterogram agreed well with established clade designations and shows that available *Es. marmotae* strains constitute a distinct species in the *Escherichia* genus (Fig. 1). Other former cryptic *Escherichia* clades recently defined as novel species also cluster by themselves, whereas a selection of diverse *Es. coli* appear together, albeit with a larger MASH distance distribution than is the case with *Es. marmotae*. The discrepancy is likely a result of strain selection bias due to a larger available pool of *Es. coli* sequences to pick from as well as a historically wide definition of what constitutes *Es. coli*.

**Human clinical strains are scattered within the *Es. marmotae* phylogeny.** To investigate the phylogenetic relationships among the 41 *Es. marmotae* genomes further (36 from GenBank and 5 from this study), we constructed a separate phylogeny with Parsnp (9) (Fig. 2). The phylogeny was associated with strain origin and presence of antimicrobial resistance genes. We also performed a Roary pan-genome analysis. Eight of the available *Es. marmotae* whole genomes turned out to represent human isolates (four from feces, two from blood, one from urine, and one with an unknown isolation source). Most of them were originally submitted as *Es. coli* but later reassigned as *Es. marmotae* by NCBI.

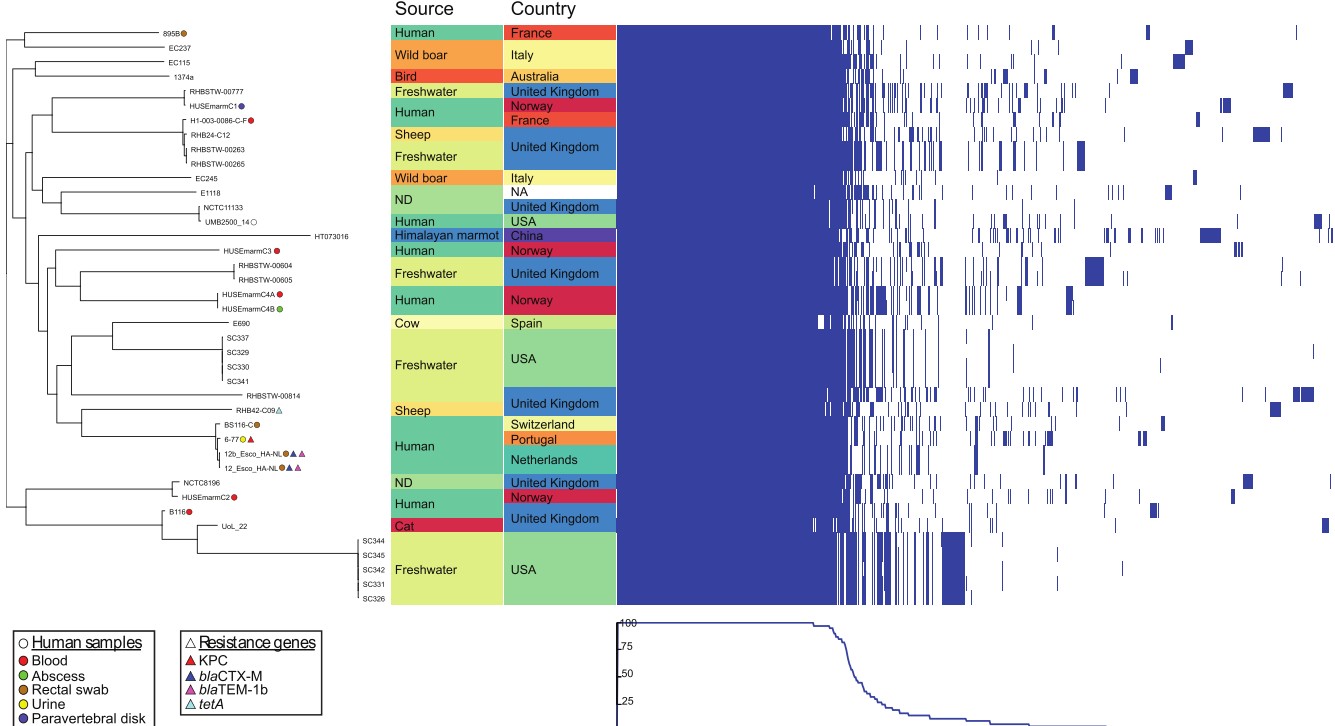

**FIG 2** Left, midpoint-rooted phylogeny of *Es. marmotae* strains, with highlighting of human samples and isolation sites in colored dots and the presence of resistance genes in colored triangles. Middle, isolate origin by location and country is given and color coded. Right, pan-genome map showing approximate relative size of the core and accessory genomes, where the blue bar represents presence of a gene. The graph in the bottom right depicts the proportion of isolates harboring a given gene.

The pan-genome of these 41 isolates included 11,549 genes, of which 3,163 (27,4%) were conserved in all strains. As these few strains were found within a broad context of environments and contained a large accessory genome, we predict *Es. marmotae* to be a generalist species like *Es. coli* with similar species ubiquity and similar core- and pan-genome features (10).

The 13 genomes from isolates of human origin were scattered throughout the phylogenetic tree among strains from environmental origins. In two cases, isolates linked to the same patient naturally appeared together, specifically HUSEmarmC4A and B, and presumably 12- and 12b-Esco-HA-NL.

**Available data show infrequent occurrences of antimicrobial resistance.** We could not identify any resistance genes in our five isolates, and all were phenotypically susceptible to the antimicrobials in our standard susceptibility panels (Table S2).

Among the other *Es. marmotae* strains included in this study, genetic resistance markers as determined by the NCBI AMRfinder database were identified in only four. Among these, three were isolated from human sources, thereby linking antimicrobial resistance in this species to human hosts. A single isolate (6-77, urinary tract infection [UTI], Portugal) harbored the *bla*KPC carbapenemase, two isolates, presumably representing the same strain (12-Esco-HA-NL and 12b-Esco-HA-NL, rectal swabs, the Netherlands), had a broad-spectrum beta-lactamase (*bla*CTX-M) in addition to *bla*TEM-1b, and a single strain (RHB43-C09) harbored a gene conferring tetracycline resistance (*tetA*). Only the latter strain did not originate from a human source but from sheep. The *tetA* gene has previously been found in over 70% of *Es. coli* sampled from ruminants (11).

The presence of beta-lactam resistance genes in strains from human sources may speculatively represent introduction by horizontal gene transfer from other microbes within the human gut. Others have found the presence of diverse beta-lactamases in *Es. marmotae*, including extended-spectrum beta-lactamase (ESBL) and carbapenemases (ESBL$_{CARBA}$), and also found extensive sharing of resistance genes between

different *Enterobacteriaceae* retrieved from the same healthy host fecal samples, indicating frequent spatiotemporal sharing of mobile genetic elements containing resistance genes (12).

**Escherichia marmotae virulence reevaluated.** The assessment of virulence in microbes normally includes searching for genes and gene clusters that encode various properties, some of which have been demonstrated to increase virulence in disease models, and others where only evidence of involvement in a host-microbe interaction is shown (13). Liu et al. have already genotypically and phenotypically characterized the virulence traits of *Es. marmotae* (2). However, their analysis included isolates from the intestines of presumably healthy *Marmota himalayana* exclusively. Using the virulence finder database (VFDB) combined with isolates from a variety of isolation sources, we obtained a broader and somewhat different genetic picture. All genes described here are found with greater than 98% homology to the database entry, as reported by Abricate.

In concurrence with Liu et al., we found that all *Es. marmotae* contain the enterotoxin-encoding gene *astA* (14). However, only the type strain (HT073016), the sole isolate from *M. himalayana* in our genome collection, contained the *efa1* gene encoding an adherence factor, the *Yersinia* T3SS effector *yopJ*, and a truncated version of the *Shigella*-associated virulence gene *ipaH*.

In addition to the virulence genes previously reported, we found several iron acquisition systems within all *Es. marmotae* genomes, notably *entA/B/C/D/E/F/S* encoding the iron siderophore enterobactin (15) and *fepA/B/C/D/G* and *fes*, which encode another siderophore ferrienterochelin (16). All strains also encoded *chuS/T/U/V/W/X* involved in heme uptake and metabolism (17, 18). These systems aid in survival during invasive infections by avoiding nutritional immunity, where host sequestration of essential nutrients prevents bacterial survival (17). They are also associated with increased virulence (19).

Further, all *Es. marmotae* contained *csgB/D/E/F/G*, which are genes facilitating production of extracellular curli proteins involved in matrixes promoting cellular adhesion and biofilm formation (20) as well as *fimA/B/C/D/E/F/G/H/I* encoding type 1 fimbriae associated with urinary tract infection (21). They also harbor *ompA*, which encodes the outer membrane protein A associated with several pathogenic properties, including evasion of host defense systems during intracellular migration in neonatal meningitis and inactivation of the complement system (22).

All *Es. marmotae* also had a type 2 secretion system associated with extracellular release of the enterotoxin produced by enterotoxigenic *Es. coli* (ETEC; *gspC/D/E/F/G/H/I/J/K/L/M*) (23). The toxin itself was not found. All also encoded *kpsD*, which mediates export of group 2 capsular polysaccharides across the outer membrane, notably the O antigen involved in bacterial survival in blood and urinary tract persistence (24). We also found *kpsM*, associated with the virulence of extra intestinal pathogenic *E. coli* (ExPEC) (25).

Finally, 11 of 41 isolates, with no particular habitat affinity, harbored *faeC/D/E/F/H/I/J* genes, which constitute an operon producing fimbriae associated with the pathogenic processes in ETEC (26).

By using the *E. coli* O-groups and H-types database (ECOH) *Es. coli in silico* serotyper (27), we found all *Es. marmotae* strains to contain the *fliC*-H56 flagellar antigen but a wide array of 12 O antigens. The H56 serotype is therefore a potential marker of *Es. marmotae* in historic data. Interestingly, the H56:O103 serotype of *Es. coli* has earlier been reported to populate the intestines of Norwegian sheep (28). HUSEmarmC2, the isolate giving pyelonephritis in one patient, was of this serotype. HUSEmarmC3, the isolate retrieved from the patient in previous contact with sheep manure, had the serotype Onovel21:H56.

**Routine identification of Es. marmotae in the clinical lab.** Although we did not systematically search for *Es. marmotae*, the four cases from our lab were discovered over a period of only 6 months. We therefore anticipate that, unlike the recently characterized species *Es. ruysiae* (former *Escherichia* clades III and IV) (13), *Es. albertii*, and *Es. fergusonii* (supplemental material in ref. 6), *Es. marmotae* could represent a relatively

regular cause of human-invasive infections. In two previous retrospective investigations of large collections of *Es. coli* blood culture isolates, only one (7) and two (29) isolates (0.1 to 0.2% of the analyzed strains), respectively, were reclassified as "*Escherichia* cryptic clade V," indicating that *Es. marmotae* only rarely causes bloodstream infections. Since we have not systematically investigated a larger number of isolates from different isolation sites, we cannot speculate more on the frequency of *Es. marmotae* as a cause of human infections. Reliable routine identification of *Es. marmotae* by clinical laboratories will be necessary to assess the prevalence of infections in humans.

*Escherichia marmotae*, or "*Escherichia* cryptic clade V," is phenotypically indistinguishable from *Es. coli*. This phenotypic homology is the historic background for the term "cryptic" *Escherichia* (30). The four clinical cases from our lab were discovered by experienced microbiologists reacting to an unusually low score for *Es. coli* by MALDI-TOF MS. At that time, no spectra for *Es. marmotae* were available in our MALDI-TOF MS database. This underscores the risk of species misclassification by MALDI-TOF MS when the correct species is not represented in the database, as previously addressed by Body et al. (12). They found that *Mycobacterium phocaicum* was regularly misidentified as the closely related *Mycobacterium mucogenicum* due to the lack of spectra for *M. phocaicum*. Unfortunately, the inclusion of a single spectrum for *Es. marmotae* (strain HWL_017a HWH) in the latest update of the Bruker Biotyper database (L2020 9607MSP) does not seem to enable robust identification and discrimination from *Es. coli*. By reanalysis using the novel database, our isolates obtained scores in the range of 1.79 to 2.02 against *Es. marmotae* versus 1.70 to 2.06 against *Es. coli* (Table S3). As for *Es. coli*, we assume it will be necessary with a wider range of spectra to cover the intraspecies variability of *Es. marmotae*. The isolates from the four patients in this study have been sent to Bruker for the generation of novel reference spectra, but these are still not available.

Available variants of the *Es. marmotae* complete 16S rRNA gene display pairwise homologies between 99.4 and 100% and form a distinct branch in phylogenetic comparisons with the other members of the *Escherichia* genus. Reliable discrimination from *Es. coli* based on the 16S rRNA V1-V3 variable areas as commonly used in diagnostic laboratories is formally possible with pairwise homologies varying from 96.6 to 99.0%. Corresponding homologies with *Es. albertii*, *Es. fergusonii*, and *Es. ruysiae* were 96.9 to 98.6%, 98.0 to 98.6%, and 97.9 to 99.0%, respectively. There appears to have been a lack of 16S rRNA references for the previous *Escherichia* cryptic clade V in GenBank prior to the description of *Es. marmotae*. This might have delayed the recognition of *Es. marmotae/Escherichia* cryptic clade V as a human pathogen.

**Conclusions.** *Escherichia marmotae* is a recently described species phenotypically indistinguishable from *Es. coli*. We have shown that it can cause invasive human infections, involving blood, bone, the urinary tract, and the bile duct system as well as septic shock. It further has the capacity to acquire broad-spectrum $\beta$-lactamases, including carbapenamases. A wide range of isolation sources combined with a large accessory genome is indicative of a true generalist bacterium, and the disease-causing strains in this study were found to scatter among environmental and animal isolates in the phylogenetic analyses. MALDI-TOF MS has the potential to provide robust routine identification, but a sufficient range of *Es. marmotae* spectra in the database is not yet available. We currently see no reason to fear that misidentifying a clinical *Es. marmotae* isolate as *Es. coli* would have any implications for the patient, but future surveillance is appropriate to verify this.

## MATERIALS AND METHODS

**Ethical statement.** The study was approved by the Regional ethical committee of Western Norway (REK 322324). Written, informed consent to use clinical data were obtained from three of the four patients. For the fourth patient, only microbiological data are provided. All authors declare no conflicts of interest.

**Isolate retrieval.** The clinical isolates in this study were retrieved in our routine diagnostic lab at Haukeland University Hospital, Bergen, Norway. The blood culture isolates were recovered in aerobe and anaerobe bottles in the BacT/Alert Virtuo (bioMérieux, Durham, NC, USA) automated blood culture system and subcultured on blood and lactose agar plates. The nonblood specimens were cultured directly on blood and lactose agar plates. All plates were incubated overnight at 35°C in a $CO_2$-enriched (5%) atmosphere. After routine diagnostics, the isolates were stored in Greaves medium at −80°C and subcultured on blood agar plates overnight as described above prior to DNA extraction.

**MALDI-TOF MS.** For matrix-assisted laser desorption ionization–time of flight mass spectrometry (MALDI-TOF MS), we used a Microflex LT mass spectrometer (Bruker Daltonics, Billerica, MA, USA) with database versions K2019 8468MSP (without *Es. marmotae* spectra) during the strain collection period and version L 2020 9607MSP (which includes a single *Es. marmotae* spectrum) from September 2021. Isolates were smeared directly to the MALDI-TOF MS target plate and analyzed according to the standard routine protocol without an on-plate formic acid treatment.

**DNA extraction.** Genomic DNA for Sanger sequencing of the partial 16S rRNA gene and Illumina whole-genome sequencing was extracted from overnight bacterial colonies. Mechanical lysis of bacterial cells was performed using the Septifast Lys kit (Roche Diagnostics, Mannheim, Germany), MagNa Pure bacteria lysis buffer (Roche), and the MagNA Lyser instrument (Roche), followed by extraction on a MagNaPure Compact instrument (Roche) using the MagNA Pure Compact nucleic acid isolation kit I (Roche) according to the manufacturer's instructions.

**Sanger sequencing of the 16S rRNA gene.** Amplification of the variable areas V1-V3 of the 16S bacterial rRNA gene was done using dual priming oligonucleotides as previously described (31), with 5′-end modifications as outlined by Dyrhovden et al. (32) (16S_DPO_short-F, 5′-AGAGTTTGATCMTGGCTCAIIIIIAACGCT-3′, and 16S_DPO_short-R, 5′-CGGCTGCTGGCAIIIAITTRGC-3′). The PCR mixture consisted of 12.5 $\mu$L of *Ex Taq* SYBR master mix (TaKaRa, Otsu, Japan), 0.4 $\mu$M of each primer, 8.5 $\mu$L of PCR-grade water, and 2 $\mu$L of template DNA. The PCR was run on a QuantStudio5 real-time PCR instrument (Thermo Fisher, Waltham, MA USA). The PCR thermal profile included an initial polymerase activation step of 10 s at 95℃, followed by 45 cycles of 10 s at 95℃ (melt), 15 s at 60℃ (annealing), and 20 s at 72℃ (extension). Sanger sequencing was performed in a core facility using an ABI 3730 DNA analyzer (Applied Biosystems/Thermo Fisher).

**Whole-genome sequencing and bioinformatic analyses.** Five isolates from four patients were sequenced on an Illumina MiSeq system with 150-bp paired-end reads. Libraries were generated using a Nextera XT DNA library preparation kit (Illumina, San Diego, CA, USA). Raw data were deposited in ENA with the accession numbers stated in Table S1 in the supplemental material. Available genomes representing the *Escherichia* genus and used in the phylogeny published by Denamur et al. (6) were downloaded as comparators (Table S1). All available complete and draft *Es. marmotae* genomes as of July 2021 were downloaded via the NCBI assembly database using "*Escherichia marmotae*" as a search word. Bioinformatic analyses were done by installing and running the Bactopia v.1.7.1 (33) environment, which contains the software and version documentation described below. Briefly, adaptor trimming was done with Trimmomatic v. 0.39, and assembly was done with a combination of Shovill v. 1.0.9se and SKESA v. 2.3.0 (34) and annotated with Prokka v. 1.14.5 (35). Assembly statistics are available in Table S4. Using Abricate v.1.0.1 (https://github.com/tseemann/abricate) with blastn v. 2.9.0, antimicrobial resistance genes were predicted with the NCBI AMRfinder database (36) with a focus on clinically relevant resistance markers. Virulence genes were predicted with Abricate using the VFDB database (37), and serotyping was done using the ECOH database (27). The *Escherichia* genus was clustered by importing the pairwise comparison matrix from MASH v. 2.2.2 (8) into R to create a heat map using a modified version of the script provided by Desmet et al. (38). A pan-genome analysis of *Es. marmotae* was done with Roary v. 3.13.0 (39), and a phylogeny was created with Parsnp v.2.1.8 (9) using -xc flags and using the type strain HT073016 as reference. Phylogeny with metadata and a pan-genome gene presence/absence scheme was presented via phandango (40). All software were run with default settings unless otherwise noted.

**Data availability.** The raw data and assemblies of the five sequenced isolates have been published under BioProject (PRJEB47670) and BioSample, and accession numbers are provided in Table S1.

## SUPPLEMENTAL MATERIAL

Supplemental material is available online only.
**SUPPLEMENTAL FILE 1**, XLSX file, 0.05 MB.
**SUPPLEMENTAL FILE 2**, XLSX file, 0 MB.
**SUPPLEMENTAL FILE 3**, XLSX file, 0.01 MB.
**SUPPLEMENTAL FILE 4**, XLSX file, 0.1 MB.

## ACKNOWLEDGMENT

We would like to thank the Department of Microbiology at Haukeland University Hospital, Bergen, for funding this study.

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
