## [Reviewer comments · Microbiology Spectrum]

Microbiology Spectrum

Escherichia marmotae - a human pathogen easily misidentified as *Escherichia coli*

Audun Sivertsen, Ruben Dyrhovden, Marit Tellevik, Torbjørn Bruvold, Eirik Nybakken, Dag Harald Skutlaberg, Ingerid Skarstein, and Øyvind Kommedal

Corresponding Author(s): Audun Sivertsen, Haukeland University Hospital

Review Timeline:

Submission Date:	October 27, 2021
Editorial Decision:	February 2, 2022
Revision Received:	February 24, 2022
Accepted:	March 15, 2022

Editor: Kevin Theis

Reviewer(s): Disclosure of reviewer identity is with reference to reviewer comments included in decision letter(s). The following individuals involved in review of your submission have agreed to reveal their identity: Erick Denamur (Reviewer #2); Irene Burckhardt (Reviewer #3)

Transaction Report:

DOI: <https://doi.org/10.1128/spectrum.02035-21>

February 2, 2022

Dr. Audun Sivertsen
Haukeland University Hospital
Department of microbiology
Jonas Lies vei 65
Bergen 5021
Norway

Re: Spectrum02035-21 (*Escherichia marmotae* - a human pathogen easily misidentified as *Escherichia coli*)

Dear Dr. Audun Sivertsen:

Thank you for submitting your manuscript to Microbiology Spectrum. The manuscript has been reviewed by three experts in the field. Based on these comments, I am requesting modification of the manuscript. When submitting the revised version of your paper, please provide (1) point-by-point responses to the issues raised by the reviewers as file type "Response to Reviewers," not in your cover letter, and (2) a PDF file that indicates the changes from the original submission (by highlighting or underlining the changes) as file type "Marked Up Manuscript - For Review Only". Please use this link to submit your revised manuscript - we strongly recommend that you submit your paper within the next 60 days or reach out to me. Detailed instructions on submitting your revised paper are below.

Link Not Available

Sincerely,

Kevin R. Theis

Journals Department
Reviewer comments:

Reviewer #1 (Comments for the Author):

General comments:

This is a very interesting study which describes the potential clinical importance of *Escherichia marmotae*, as the strains studied were isolated from the lesion sites or blood from diseased patients. Prior to this, the potential pathogenicity of *E. marmotae* was presumed through genomic prediction and/or in-vitro assays.

I find very interesting the way authors use point by point conclusions as sub-headers for the Results & Discussion section.

Nevertheless, the paper has not been organized properly and readers need to go back and forth through the MS to have a clear picture of the study. I understand that Microbiology Spectrum has a format-neutral submission policy, but I believe that including the Methods section before the Results and Discussion might ease comprehension. Furthermore, I consider that some parts

need to be elaborated in more detail. For example, the M&M section lacks important details such as isolation methods, bioinformatic tools versions, etc. The results section needs to include bioinformatic summary of sequencing output and assembly performance, etc.

It really helps to have line and page numbers for reviewers to comment on typos or make specific comments, and this observation is clearly stated in the journal submissions guidelines (<https://journals.asm.org/format-neutral-submissions>). I added the line numbers to help with the reviewing process (L1 - We hereby present....) and will refer my comments to those line numbers.

Specific comments:

ABSTRACT

L9, "The invasive isolates were scattered among isolates from a range of non-human sources": The type of analysis used to reach this conclusion should be briefly stated, otherwise it is not understood here.

INTRODUCTION:

L1: typo in *Escherichia*

L24: Clade 5 vs. clade V in abstract. Please check throughout the manuscript and use consistently one way or the other, clade V is the term most widely used.

L33: authors claim to present 4 case reports. However, the manuscript does not actually fulfil a case report by definition. The journal has reporting guidelines for case reports in case authors need clarification: <https://journals.asm.org/reporting-guidelines>. Maybe authors can re-phrase "Here, we present four case reports where *E. marmotae* was isolated as the likely pathogen of invasive human infections" as follows: Here, we report four human clinical cases where *E. marmotae* was isolated as the likely case of invasive infections. Also, in the sub-heading in L55 "case reports" could be deleted.

RESULTS & DISCUSSION:

L71 & L80: Please, rephrase or define the meaning of "two sets of blood cultures".

L113, "Six of the available *E. marmotae* whole genomes...": Is this an error? Both in Fig. 2 and in suppl. T1, there are 8 human isolates in addition to the 5 described in this study. Maybe you mean that 6 of those 8 available genomes were originally submitted as *E. coli* but later reassigned as *E. marmotae*? Please, clarify. Same in L122, if you consider all human isolates, it would be 13 rather than 11.

L133-143: Did authors try to use other AMR databases for comparison purposes? Some of the strains downloaded from other studies do have AMR determinants as defined with other databases besides those mentioned here. Please check in respective publications. Once this is revised, re-evaluate the statement in L111 "The phylogeny was associated with (strain origin and) presence of antimicrobial resistance genes" just in case this wouldn't hold true.

L146: how can authors state that these accessory genes are reminiscent from *E. coli*? Did authors conduct any genome comparative analyses such as mauve?

L147: same here. Did authors try to predict mobile genetic elements from the isolates? What is a "visual correlation"? Correlations cannot be "visually" inferred.

CONCLUSIONS: Maybe the authors could briefly discuss on the clinical implications of misidentifying *E. marmotae* as *E. coli*.

METHODS:

L279: Can authors provide more details about the microbiological isolation of the *E. marmotae* strains? Also, how were the isolates stored since isolation and how were they recovered prior to sequencing?

L304-305: No need to state here that sequences were deposited in ENA; already in "Data Availability". However, it would be reasonable to briefly describe how authors selected the other *E. marmotae* strains (source of information, key words, etc).

L309: can authors provide the summary of descriptive statistics regarding the sequencing output and assembly assessment (total reads per sample, Q score coverage, N50, number of contigs, GC content, size of draft genome, etc) to the results section?

Did authors pre-processed raw data before the Bactopia pipeline? Adapters or barcode trimming in case of multiplexing, etc?

I am not aware about Bactopia using abricate to screen for antimicrobial resistance. Did authors use abricate independently from Bactopia? If so, can authors please elaborate more? Which versions of blastn and abricate were used? Did authors screen for antimicrobial point mutations?

Please provide the versions of all bioinformatic tools used and the date of the last update of the databases.

L319: use "default settings" rather than "standard settings".

Figure 2: If no country data is available for isolate E1118, authors could consider NA instead of white/blank.

SupT2: Are there any differences between those cases designated as "s" and "S" for a particular antimicrobial? If so, please indicate in a footnote. Also, consider providing cut-off values rather than referring to the Nordicast guidelines of September 2021.

Cell A13: typo in Trimethoprim

REFERENCES: Please thoroughly check for formats, for example italics in species names.

Some minor suggestions:

- Use the past tense to describe the results.
- Abbreviate to *E. marmotae* after first description in full.
- 16 rRNA should always be followed by "gene" as in "16 rRNA gene".
- The authors seem to include in the term "environmental isolates" all non-human isolates. However, I would recommend splitting environmental isolates into isolates of animal and environmental origin.

Reviewer #2 (Comments for the Author):

The paper of Sivertsen et al. describes four cases of extra-intestinal infections in human due to *Escherichia marmotae*. From the complete genome sequences, the authors suggest that *E. marmotae* has a high virulence potential. The topic is of interest, the methods are accurate but the conclusions are overstated and should be mitigated for the following reasons.

First, most of the cases occurred in immunosuppressed patients.

Second, *E. marmotae* is not so frequent in bloodstream infections as large series for example from England (see Kallonen et al., *Genome Res*, 2017) and France (see Royer et al., *Genom Med* 2021) found it at a very low rate (less than 0.1%).

Third, the list of the "virulence genes" presented is not convincing: *ent*, *fep* and *fim* operons as well as *ompA* are found in *E. coli* K-12, an archetypal non-virulent strain.

Fourth, when tested in a mouse model of sepsis representative of the intrinsic virulence of the strains, *E. marmotae* do not kill mice (see ref 18).

Reviewer #3 (Comments for the Author):

The authors present an interesting study on the prevalence of the recently described species *E. marmotae*.

It describes four different clinical manifestations. Here are some suggestions for an addendum to the manuscript.

1. As mentioned there is only one MALDI-TOF spectrum in the database. Did the authors try to add mean spectra (MSP) of their own strains to the database (normally the manufacturers help with that, either with a protocol or with actually providing the spectra from the strains)? It would be interesting what a re-analysis of the *E. coli* spectra would result in!
2. Apparently *E. marmotae* was already found in Norwegian sheep. Are there any information on what it causes there?
3. Where all patients asked for their interaction with sheep?
4. The last sentence of the introduction is irritating. It is too common place. Either omit it or explain in detail why you think it will add to the monitoring apart from "we have a new species". From what I read *E. marmotae* is a gram-negative bacterium with no overtly terrible resistance markers and was successfully treated 4 times?!
5. Did you look into the biochemistry of the bacteria? Would it have been possible to identify it with biochemical means like gram-negative cartridges e.g.?
6. supplementary data on susceptibility testing: what is the difference in meaning of s and S?

Staff Comments:

Preparing Revision Guidelines

To submit your modified manuscript, log onto the eJP submission site at <https://spectrum.msubmit.net/cgi-bin/main.plex>. Go to Author Tasks and click the appropriate manuscript title to begin the revision process. The information that you entered when you first submitted the paper will be displayed. Please update the information as necessary. Here are a few examples of required

updates that authors must address:

Please return the manuscript within 60 days; if you cannot complete the modification within this time period, please contact me. If you do not wish to modify the manuscript and prefer to submit it to another journal, please notify me of your decision immediately so that the manuscript may be formally withdrawn from consideration by Microbiology Spectrum.

Response to reviewers

To the editor

We sincerely thank you and the reviewers for a thorough review process of our paper resulting in a more precise and cohesive work. We have implemented most of the suggestions as described in the point-by-point response.

We would also like to mention, that since we finished the first version of this manuscript, we have encountered an additional nine *Escherichia marmotae* strains, one from blood and eight from urine specimens. We run a medium sized lab, serving approximately 500.000 people. Although these latter strains were discovered too late to be included in this article, they serve to indicate that *E. marmotae* indeed is present if looked for.

Reviewer comments:

Reviewer #1

(Comments for the Author):

General comments:

This is a very interesting study which describes the potential clinical importance of *Escherichia marmotae*, as the strains studied were isolated from the lesion sites or blood from diseased patients. Prior to this, the potential pathogenicity of *E. marmotae* was presumed through genomic prediction and/or in-vitro assays.

I find very interesting the way authors use point by point conclusions as sub-headers for the Results & Discussion section.

Nevertheless, the paper has not been organized properly and readers need to go back and forth through the MS to have a clear picture of the study. I understand that Microbiology Spectrum has a format-neutral submission policy, but I believe that

including the Methods section before the Results and Discussion might ease comprehension. Furthermore, I consider that some parts need to be elaborated in more detail. For example, the M&M section lacks important details such as isolation methods, bioinformatic tools versions, etc. The results section needs to include bioinformatic summary of sequencing output and assembly performance, etc.

It really helps to have line and page numbers for reviewers to comment on typos or make specific comments, and this observation is clearly stated in the journal submissions guidelines (<https://journals.asm.org/format-neutral-submissions>). I added the line numbers to help with the reviewing process (L1 - We hereby present....) and will refer my comments to those line numbers.

Response:

We thank you for your positive and constructive remarks. To accommodate a better flow in the text, we have now placed the Materials & Methods section between the Introduction and the Results/Discussion section. We have also included isolation methods (routine isolation from the clinical lab) (lines 62-70) and bioinformatic tool versions. We have also provided a summary of the Sequencing output data in the new supplementary table S4.

Line numbers have been added - we will refer to line numbers below to track changes in the manuscript text.

Specific comments:

ABSTRACT

L9, "The invasive isolates were scattered among isolates from a range of non-human sources": The type of analysis used to reach this conclusion should be briefly stated, otherwise it is not understood here.

Response: The sentence now reads "The invasive isolates were scattered among isolates from a range of non-human sources in the phylogenetic analyses, thus indicating inherent virulence in multiple lineages." (Lines 16 to 18)

INTRODUCTION:

L1: typo in Escherichia

Response: Corrected.

L24: Clade 5 vs. clade V in abstract. Please check throughout the manuscript and use consistently one way or the other, clade V is the term most widely used.

Response: Clade designation has been corrected to "Clade V" throughout the manuscript.

L33: authors claim to present 4 case reports. However, the manuscript does not actually fulfil a case report by definition. The journal has reporting guidelines for case reports in case authors need clarification: <https://journals.asm.org/reporting->

guidelines. Maybe authors can re-phrase "Here, we present four case reports where *E. marmotae* was isolated as the likely pathogen of invasive human infections" as follows: Here, we report four human clinical cases where *E. marmotae* was isolated as the likely case of invasive infections. Also, in the sub-heading in L55 "case reports" could be deleted.

Response: We thank you for these suggestions, which have both been implemented (Lines 48 and 136-137).

RESULTS & DISCUSSION:

L71 & L80: Please, rephrase or define the meaning of "two sets of blood cultures".

Response: We have rephrased this in the case descriptions (Lines 150 to 151 and 158).

L113, "Six of the available *E. marmotae* whole genomes...": Is this an error? Both in Fig. 2 and in suppl. T1, there are 8 human isolates in addition to the 5 described in this study. Maybe you mean that 6 of those 8 available genomes were originally submitted as *E. coli* but later reassigned as *E. marmotae*? Please, clarify. Same in L122, if you consider all human isolates, it would be 13 rather than 11.

Response: Thank you for pointing this out. This was an error from our side. The correct number is 8 human strains, including four faecal strains, two from blood, one from urine, and one (UMB2500_14) with unknown isolation source. We have corrected the numbers and modified the sentence regarding species reassignment by NCBI (Lines 187 to 190).

L133-143: Did authors try to use other AMR databases for comparison purposes? Some of the strains downloaded from other studies do have AMR determinants as defined with other databases besides those mentioned here. Please check in respective publications. Once this is revised, re-evaluate the statement in L111 "The phylogeny was associated with (strain origin and) presence of antimicrobial resistance genes" just in case this wouldn't hold true.

*Response: We expanded our search to include the following AMR databases which were downloaded with abricate: amrfinder, card, megares, argannot, in addition to AMRfinder+. Several of the databases reported the presence of 30-50 resistance associated genes, of which none were deemed as clinically relevant apart from the ones already mentioned in the text. The remaining suggested resistance determinants were chromosomal genes present in most/all *E. marmotae* genomes, and may represent resistance towards metals and pesticides not relevant in the clinical setting. We have therefore chosen not to include them in the text, but define our focus on line 115.*

L146: how can authors state that these accessory genes are reminiscent from *E. coli*? Did authors conduct any genome comparative analyses such as mauve?

Response: In retrospect we see that this was a poor statement that should never have been included. We wanted to say something about the pan genome plot to the right in

the phandango figure (Fig 2), but the strain collection is too diverse and too small to draw any conclusions. The sentences have been deleted.

L147: same here. Did authors try to predict mobile genetic elements from the isolates? What is a "visual correlation"? Correlations cannot be "visually" inferred.

Response: These sentences have been removed, please see above.

CONCLUSIONS: Maybe the authors could briefly discuss on the clinical implications of misidentifying *E. marmotae* as *E. coli*.

Response: A sentence has been added to the conclusions chapter (Lines 321 to 323)

METHODS:

L279: Can authors provide more details about the microbiological isolation of the *E. marmotae* strains? Also, how were the isolates stored since isolation and how were they recovered prior to sequencing?

Response: This information has now been provided (Lines 62 to 70)

L304-305: No need to state here that sequences were deposited in ENA; already in "Data Availability". However, it would be reasonable to briefly describe how authors selected the other *E. marmotae* strains (source of information, key words, etc).

Response: the following has now been included: "All available complete and draft E. marmotae genomes as of july 2021 were downloaded via the NCBI assembly database using "Eschericha marmotae" as a search word." (Lines 106 to 108)

L309: can authors provide the summary of descriptive statistics regarding the sequencing output and assembly assessment (total reads per sample, Q score coverage, N50, number of contigs, GC content, size of draft genome, etc) to the results section?

Response: A new supplemental table S4 has been made, describing the above parameters, and mentioned in the text (Line 113).

Did authors pre-processed raw data before the Bactopia pipeline? Adapters or barcode trimming in case of multiplexing, etc?

Response: Trimming was done with trimmomatic, this has now been added to Materials & Methods (Line 111).

I am not aware about Bactopia using abricate to screen for antimicrobial resistance. Did authors use abricate independently from Bactopia? If so, can authors please elaborate more? Which versions of blastn and abricate were used? Did authors screen for antimicrobial point mutations?

Response: Bactopia takes use of Ariba which analyzes content of genes of interest from raw data - albeit, despite trying to install Ariba within the Bactopia framework on both a linux system, an intel Mac and an M1 Mac, we could not get it to work. We thus

independently used Abricate - version numbers are given in "WGS and bioinformatic analyses"-chapter (lines 100 to 124).

Please provide the versions of all bioinformatic tools used and the date of the last update of the databases.

Response: Versions have now been provided. The E. marmotae draft and whole genomes were those available in GenBank as of July 2021 (Line 107).

L319: use "default settings" rather than "standard settings".

Response: The sentence has been corrected (Line 124).

Figure 2: If no country data is available for isolate E1118, authors could consider NA instead of white/blank.

Response: "NA" has been added.

SupT2: Are there any differences between those cases designated as "s" and "S" for a particular antimicrobial? If so, please indicate in a footnote. Also, consider providing cut-off values rather than referring to the Nordicast guidelines of September 2021.

Response: Corrected. This was an unintended distinction, s and S means the same.

Cell A13: typo in Trimethoprim

Response: Corrected.

REFERENCES: Please thoroughly check for formats, for example italics in species names.

Response: Corrections have been performed.

Some minor suggestions:

- Use the past tense to describe the results.
- Abbreviate to *E. marmotae* after first description in full.
- 16 rRNA should always be followed by "gene" as in "16 rRNA gene".
- The authors seem to include in the term "environmental isolates" all non-human isolates. However, I would recommend splitting environmental isolates into isolates of animal and environmental origin.

Response: Thank you for suggestions. Corrections have been made accordingly throughout the manuscript. After the first description, Escherichia marmotae has been truncated to E. marmotae except from in the start of a paragraph.

Reviewer #2 (Comments for the Author):

The paper of Sivertsen et al. describes four cases of extra-intestinal infections in human due to *Escherichia marmotae*. From the complete genome sequences, the authors suggest that *E. marmotae* has a high virulence potential. The topic is of interest, the methods are accurate but the conclusions are overstated and should be mitigated for the following reasons.

First, most of the cases occurred in immunosuppressed patients.

Second, *E. marmotae* is not so frequent in bloodstream infections as large series for example from England (see Kallonen et al., *Genome Res*, 2017) and France (see Royer et al., *Genom Med* 2021) found it at a very low rate (less than 0.1%).

Third, the list of the "virulence genes" presented is not convincing: *ent*, *fep* and *fim* operons as well as *ompA* are found in *E. coli* K-12, an archetypal non-virulent strain.

Fourth, when tested in a mouse model of sepsis representative of the intrinsic virulence of the strains, *E. marmotae* do not kill mice (see ref 18).

Response: All our patients suffered severe invasive infections. As the reviewer comments, one patient had haematologic cancer, leading to severe immunosuppression. Two others had diabetes type 2 and high age respectively, which may lead to mild immunosuppression, but these are very common conditions which also increase the risk of infection with E. coli.

We appreciate the references provided by the reviewer and two of these have now been implemented and discussed in the text (Lines 274 to 278). The third reference, Kallonen et al. (Genome Res 2017) only included E. coli strains from phylogroups A-F and i.e. none of the cryptic clades/species II->V. Cryptic clades are not mentioned in this work.

Among our five isolates, three were from other sources than blood. The publications from Royer and Clermont included almost only blood culture isolates and therefore do not necessarily reflect the overall prevalence in human clinical samples.

We have modified the conclusion somewhat and also included a sentence regarding clinical consequences of misidentification as requested by reviewer 1. In addition, we have included a discussion regarding the prevalence of E. marmotae in clinical samples in the section "Routine identification of E. marmotae in the clinical lab" (Lines 274 to 280)

Reviewer #3 (Comments for the Author):

The authors present an interesting study on the prevalence of the recently described species *E. marmotae*.

It describes four different clinical manifestations. Here are some suggestions for an addendum to the manuscript.

1. As mentioned there is only one MALDI-TOF spectrum in the database. Did the authors try to add mean spectra (MSP) of their own strains to the database (normally the manufacturers help with that, either with a protocol or with actually providing the

spectra from the strains)? It would be interesting what a re-analysis of the *E. coli* spectra would result in!

Response: Our lab does not currently have the certification required to add spectra to the MALDI-ToF database, albeit all strains from this study have been sent to Bruker for future incorporation of spectra (they are still not available). This is now mentioned in the text (Lines 298 to 300)

2. Apparently *E. marmotae* was already found in Norwegian sheep. Are there any information on what it causes there?

Response: We have no information suggesting anything other than E. marmotae being a gut commensal in ruminants. We also want to add that past reports of this species are scarce, and literature research is hampered by E. marmotae not being considered a separate species until recently.

3. Where all patients asked for their interaction with sheep?

Response: We did not systematically ask about interaction with sheep, but we did ask for contact with household animals. Patients 1 and 4 reported no such contact and for patient 2 this information was not available as he/she did not consent to participate.

4. The last sentence of the introduction is irritating. It is too common place. Either omit it or explain in detail why you think it will add to the monitoring apart from "we have a new species". From what I read *E. marmotae* is a gram-negative bacterium with no overtly terrible resistance markers and was successfully treated 4 times?!

Response: The sentence has been deleted.

5. Did you look into the biochemistry of the bacteria? Would it have been possible to identify it with biochemical means like gram-negative cartridges e.g.?

Response: Available literature suggests that E. marmotae is indistinguishable from E. coli using phenotypic methods. We have now included a reference to support this (Line 285)

6. supplementary data on susceptibility testing: what is the difference in meaning of s and S?

Response: This was an unintended typo. There is no difference. The issue has now been corrected.

March 15, 2022

Dr. Audun Sivertsen
Haukeland University Hospital
Department of microbiology
Jonas Lies vei 65
Bergen 5021
Norway

Re: Spectrum02035-21R1 (*Escherichia marmotae* - a human pathogen easily misidentified as *Escherichia coli*)

Dear Dr. Audun Sivertsen:

Your manuscript has been accepted, and I am forwarding it to the ASM Journals Department for publication. You will be notified when your proofs are ready to be viewed.

Sincerely,

Kevin R. Theis
Editor, Microbiology Spectrum

Journals Department
Supplemental Table 3: Accept
Supplemental table 4: Accept
Supplemental table 1: Accept
Supplemental Table 2: Accept